# Machine-Learning Approach for Automatic Detection of Wild Beluga Whales from Hand-Held Camera Pictures

**DOI:** 10.3390/s22114107

**Published:** 2022-05-28

**Authors:** Voncarlos M. Araújo, Ankita Shukla, Clément Chion, Sébastien Gambs, Robert Michaud

**Affiliations:** 1Département des Sciences Naturelles, Université du Québec en Outaouais, Ripon, QC J0V 1V0, Canada; clement.chion@uqo.ca; 2School of Arts, Media and Engineering, Arizona State University, Tempe, AZ 85281, USA; ashukl20@asu.edu; 3Département d’Informatique, Université du Québec à Montréal (UQAM), Montreal, QC H2L 2C4, Canada; gambs.sebastien@uqam.ca; 4Groupe de Recherche et d’Éducation sur les Mammifères Marins (GREMM), Tadoussac, QC G0T 2A0, Canada; rmichaud@gremm.org

**Keywords:** ocean protection, beluga whale monitoring, automatic object detection, deep learning

## Abstract

A key aspect of ocean protection consists in estimating the abundance of marine mammal population density within their habitat, which is usually accomplished using visual inspection and cameras from line-transect ships, small boats, and aircraft. However, marine mammal observation through vessel surveys requires significant workforce resources, including for the post-processing of pictures, and is further challenged due to animal bodies being partially hidden underwater, small-scale object size, occlusion among objects, and distracter objects (e.g., waves, sun glare, etc.). To relieve the human expert’s workload while improving the observation accuracy, we propose a novel system for automating the detection of beluga whales (*Delphinapterus leucas*) in the wild from pictures. Our system relies on a dataset named Beluga-5k, containing more than 5.5 thousand pictures of belugas. First, to improve the dataset’s annotation, we have designed a semi-manual strategy for annotating candidates in images with single (i.e., one beluga) and multiple (i.e., two or more belugas) candidate subjects efficiently. Second, we have studied the performance of three off-the-shelf object-detection algorithms, namely, Mask-RCNN, SSD, and YOLO v3-Tiny, on the Beluga-5k dataset. Afterward, we have set YOLO v3-Tiny as the detector, integrating single- and multiple-individual images into the model training. Our fine-tuned CNN-backbone detector trained with semi-manual annotations is able to detect belugas despite the presence of distracter objects with high accuracy (i.e., 97.05 mAP@0.5). Finally, our proposed method is able to detect overlapped/occluded multiple individuals in images (beluga whales that swim in groups). For instance, it is able to detect 688 out of 706 belugas encountered in 200 multiple images, achieving 98.29% precision and 99.14% recall.

## 1. Introduction

Most coastal and estuary environments have experienced growing industrialization over the past decades due to the increase in infrastructure construction (e.g., ports, cross-sea bridges, etc.), marine transportation, and fisheries [1]. Many of the activities associated with these industries affect the behavior and spatial distributions of marine mammals. Therefore, researchers have developed monitoring techniques. In addition, at present, marine projects are instructed to monitor their operational area for the marine mammal’s presence so that mitigation efforts can be applied [2,3]. Marine mammal observers (MMOs) are trained and instructed to monitor the sea surface for marine animals such as whales [4]. However, visual detection occurs in an instant and is subject to human biases. Thus, it is a hard task to confirm or review a detection at a later phase. The effectiveness with which an MMO can visually detect an animal is reduced by animals grouped in herds and weather conditions such as fog, rain, high sea state, or sun glare [5,6].

In recent years, monitoring marine animals has been increasingly interested in using other technologies to overcome the limitations of visual monitoring. In particular, the use of photo-identification (Photo-ID) during monitoring for mitigation purposes has increased, with some national guidelines encouraging its use and industry efforts focusing on improvising Photo-ID capabilities [7,8]. Photo-ID is a non-invasive method to monitor and identify endangered animals using camera traps [9,10]. Nevertheless, the managing and analysis of these Photo-ID data require manual efforts that include labeling and categorizing images, creating identification metadata for each photo, documenting information in databases, as well as the regular publication of reference materials [11]. Moreover, these tasks are commonly performed best by those who are intimately knowledgeable about the unique physical characteristics and social patterns of individuals in this population. In fact, this requires an exceptional level of expertise and an amount of time that may be reduced by taking advantage of developing automated technologies.

Computer vision-based approaches have increasingly been employed in marine ecology analyses [12]; among them, thermal infrared [13,14], aerial images [15], and images from boats [16] are used to monitor, detect, and identify marine animals in the wild. For example, Mejias et al. [17] described two algorithms to estimate marine populations from aerial imagery. The first algorithm used morphological operations and combined color analysis for blob detection, while the second one relies on a shape-profiling method based on the saturation channel from an HSV color space. The experiments were conducted over a set of 100 pictures captured at 1000 feet, with the two algorithms returning results with a recall of 51.4% and 48.57%, respectively. Multi-camera setups provide an opportunity to record the behavior of captive dolphins in Karnowski et al. [18], and were used to create a massive dataset providing 43,200 frames for training and 39,600 for testing to detect bottlenose dolphins. Two background subtraction methods were investigated by the authors: Robust principal component analysis (RPCA) and the mixture-of-Gaussians (MOG) approach. The reported dolphin detection performance metrics were 75.7% recall, 78.8% precision, and 77.2% F1-score using RPCA, and 76.7% recall, 63.3% precision, and 69.4% F1-score for the MOG approach.

As background subtraction is used for detection, a challenge for detection occurs when grouped animals appear at the same time. A specific convolutional neural network (CNN) named R-CNN and its extensions have been used for object detection to extract the animal from images taken in the wild and to actively ignore background noise. For example, Park et al. [19] evaluated R-CNN [20], Fast R-CNN [21], and Faster R-CNN [22] methods to detect seals and dolphins in underwater image frames. Pedersen et al. [23] applied CNN-based detector models for a marine animals dataset (fish, crabs, and starfish). Their dataset was composed of 14,518 frames with 25,613 annotations of the 6 classes: big fish, small fish, crab, jellyfish, shrimp, and starfish. Two state-of-the-art CNNs (YOLOv2 and YOLOv3) have been fine-tuned on the proposed dataset. The YOLOv2 object was pre-trained on Imagenet [23] and fine-tuned to fish datasets, and it was obtained from the VIAME toolkit [24]. The YOLOv3 detector was the original version pretrained on the Open Images dataset [25]. The fine-tuned YOLOv3 network achieved the best performance, with an mAP (mean average precision) of 39% and an mAP@50 of 84%. Most methods have been shown to work well on standard datasets such as COCO [26] and Imagenet [23]. Similarly, Borowicz et al. [13] and Guirado et al. [14] have built fine-tuned convolutional neural networks for the automated detection of whales. In Guirado et al. [14], the detection model was built using the last version of the GoogleNet Inception v3 CNN architecture, pretrained on the ImageNet database. The results were evaluated on Google Earth images in ten global whale-watching hotspots, achieving a performance F1-score of 0.81. Borowicz et al. [13] present a semi-automated pipeline for whale detection from very-high-resolution (sub-meter) satellite imagery. ResNet and DenseNet CNNs were trained and tested for each model on 31 cm-resolution imagery obtained from the WorldView-3 sensor. The model (ResNet-152) correctly classified 100% of tiles with whales and achieved an F1-score of 0.96.

These previous works have designed automatic detection systems for marine individuals such as humpback, killer, and sperm whales, as well as dolphins. To the best of our knowledge, only Harasyn et al. [27] have explored the automatic detection of beluga whales. In particular, the authors have applied a YOLOv4 convolutional neural network model to detect belugas in oblique drone imagery. The dataset used contains 721 images, with 75% of them (541 images) being used for model training, and 25% of them (180 images) for validation. This system achieves 74% precision, 72% recall, and an F1-score of 0.73. According to the authors, the proposed method is more consistent than human annotators, but has a high rate of misclassification. In particular, overlapped/occluded belugas are often identified as a single beluga, and small-scale belugas in the sea background are being missed, which results in false negatives. In addition, this work mentions that an important human effort would be required to annotate a larger training set in order to achieve a higher-accuracy system.

The detection approach is considered a fundamental step for monitoring marine mammals. In this context, we are proposing an automatic beluga whale (*Delphinapterus leucas*) detection system using photos of animals captured at sea. Detecting beluga whales through hand-held camera pictures has to face several challenges:Shortage. Belugas are infrequently seen by humans and only emerge on the water surface for a few seconds at a time;Distracter objects. Objects, such as waves, shadows, foam, and sun glare, can be visually similar to belugas, and have to be distinguished to reduce the false-positive detection rate. More precisely, these objects are regarded as distracter samples (or false positives), as seen in Figure 1;Multiple objects. Belugas swim in groups and stay close to each other on the scene, making detection difficult when more than one beluga appears overlapped or occluded (see Figure 1i);Partially visible. Mostly, a beluga’s body can be observed only partially (see Figure 1d);Image orientations. Beluga postures or behaviors captured in a snapshot are quite variable since different parts of bodies can be emerged or submerged (e.g., blowing, logging, etc.) (see Figure 1g);Low efficiency in data annotation. The annotation of belugas on long-term captured images requires checking each image and drawing boxes around belugas in a wide sea view, which is slow and requires intensive human labor.

The outline of the paper is as follows. First, Section 2 outlines the dataset details and the proposed approach using off-the-shelf detection algorithms: Mask-RCNN, SSD, and YOLO v3-Tiny. Afterwards, the experimental results are presented in Section 3. Finally, Section 4 discusses the limits of our approach and Section 5 concludes with the description of future works.

## 2. Materials and Methods

### 2.1. Data Collection

Our dataset is based on individual photo-identification boat surveys that were conducted from June to October between 1989 and 2007 as part of an ongoing long-term study of the social organization of the St. Lawrence Estuary beluga population in Québec (Canada). The Photo-ID survey is mostly conducted within the critical habitat of the St. Lawrence Estuary Beluga population and the Saguenay–St. Lawrence Marine Park, as illustrated in Figure 2.

The choice of the survey area within belugas’ summer habitat on a given day was selected in a way to avoid re-sampling areas that were covered on previous days and also according to weather conditions. This resulted in approximately 1–4 sectors and 2–5 herds observed on each survey, with about 2% of the individuals captured in two different herds on the same day. When belugas were encountered, the GPS position of the research vessel was noted, and a herd follow was undertaken to photograph as many individuals as possible within the herd using a hand-held camera. A herd follow is limited to 3 hours maximum, with the GPS location noted at least every 30 min. A detailed description of the Photo-ID survey protocol is available in Michaud [8]. Each photo was treated using standard processes for image selection, scoring, and matching [11].

### 2.2. Proposed Approach

We first curated a dataset called Beluga-5k (Section 2.3) before designing a beluga whale-detection system based on state-of-the-art deep learning detection algorithms. More precisely, we have trained a baseline detector using manually annotated images (Section 2.4). In addition, to improve the annotation performance, we have developed a semi-manual annotation strategy for candidate belugas to be retrieved and labeled by the baseline detector before refining these labels by a human annotator. The annotated images are used as input to train the detection model. Then, we have evaluated and compared three off-the-shelf object detection models: Mask-RCNN [20], SSD [28], and YOLO v3-Tiny [25] (Section 2.5). To reduce the number of false positives, the models have been fine-tuned on the proposed Beluga-5k dataset. Only fully connected layers for each backbone’s model are updated in fine-tuning, as we have frozen the initial backbone layers to use the pre-training weights from the COCO dataset [26]. Once the model is trained, as illustrated in Figure 3, a test image (single or multiple) is submitted to the detection model to verify that there is one, zero, or more than one beluga in the image. If it exists, the model detects the beluga and applies a crop to the image as output. Moreover, if there is more than one beluga in a picture, a separation (crop) is made regarding each detected object. Finally, the system has been assessed on YOLO v3-Tiny, considering both accuracy and speed as performance metrics.

### 2.3. Beluga-5k

Except for the first subset of images, which are manually annotated, we use the semi-manual beluga box annotation for single- and multiple-individual images, totaling 5122 annotated images. Table 1 summarizes the distributions of this dataset, namely, the number of individuals, training, validation, and testing images available for single and multiple individuals. For example, Figure 4 displays both single- and multiple-individual image samples.

### 2.4. Annotation

To improve the annotation efficiency, we separate the annotation process into two phases: manual and semi-manual box annotation. Specifically, we randomly select images from the Beluga-5k dataset, manually go through a few selected single-individual images, and annotate whale beluga boxes for the initial phase. Afterwards, we use the first annotated boxes to train a baseline beluga detector to get a base beluga prediction, and manually correct wrong boxes or add missing boxes.

Manual beluga box annotation. For the first phase, we manually go through the Beluga-5k dataset and select 600 images containing single belugas. Each image is annotated using the Make Sense Annotator toolkit [29]. Then, we randomly split 600 images into 500 and 100 images for training/validation purposes. Finally, we train a baseline beluga detector on this initial subset for the second annotation stage.

Semi-manual beluga box annotation. We design a semi-manual beluga box annotation strategy to improve the efficiency of annotation. This strategy involves machine-learning predictions as well as refinements of the predictions by a human. Specifically, we first apply the baseline beluga detector on single- and multiple-individual images to generate candidate boxes. These candidates can either be true positives (i.e., belugas) or distracter objects (e.g., waves, sun glare, foam, etc.). In addition, the problem of missing true positives (belugas) also exists. To address this, a human expert manually goes through predicted results, fixing labels of false positives while also correcting missing true positives. This strategy enables us to annotate single- and multiple-individual images quickly while fixing wrong annotations that result from distracter objects.

### 2.5. Object-Detection Algorithms

Generic object-detection algorithms aim at locating and detecting existing objects in pictures and labeling them [30]. In this work, we have adapted off-the-shelf general object-detection algorithms for beluga detection. More specifically, we have fine-tuned three popular object detectors, namely, Mask-RCNN, SSD, and YOLO v3-Tiny, on the Beluga-5k dataset.

Mask-RCNN [20] can be used for instance segmentation (i.e., associating specific image pixels to the detected object) and is composed of two phases. The first phase is the region proposal network (RPN). It proposes bounding box candidates regardless of object categories. The second phase is named the R-CNN stage, and consists in extracting features using RoIAlign for each proposal before performing proposal classification, bounding box regression, and mask predicting. Mask-RCNN extends the Faster RCNN [22] model with three important modifications. First, Mask-RCNN improves ROIAlign by replacing the imprecise ROI-Pooling operation used in Faster RCNN with an operation that allows accurate instance segmentation masks. Second, Mask-RCNN adds a small fully connected layer in the network head to produce the desired instance segmentation. Finally, mask and class predictions are detached, the mask being predicted by the mask network head independently from the network head, which predicts the class.

Mask-RCNN is built on a backbone convolutional neural network architecture for feature extraction [31,32]. In principle, the backbone network could be any convolutional neural network designed for image analysis. In this study, we have instantiated it with a ResNet-50 network [33]. A more recent work [20] has shown that using a feature pyramid network (FPN) based on ResNet networks as the Mask-RCNN backbone leads to improvements in both accuracy and speed. An FPN takes the benefit of the inherently hierarchical and multi-scale nature of convolutional neural networks to extract useful features at different scales for object detection, semantic segmentation, and instance segmentation.

The SSD (single shot multibox detector) algorithm [28] is based on the regression method for detection, and combines positioning and classification into one network. Relying on the VGG-16 network design [34], the SSD algorithm extracts multiple sets of feature layers in the shape of a pyramid for object class prediction and object frame labeling. In our work, we have replaced the VGG network, which is now obsolete, with the ResNet-50 backbone. Compared with the Mask-RCNN, the SSD algorithm is a multi-object-detection algorithm that instantly predicts the coordinates of the bounding box and object class. Due to the simultaneous detection in several pyramid-shaped feature maps, the time consumption performance of the single feature layer detection algorithm is eliminated, resulting in a better outcome. As a consequence, it can accurately detect task objects at different scales with a faster speed than Mask-RCNN.

The YOLO (You Only Look Once) algorithm [35] was proposed by Joseph Redmon and Ross Girshick. It operates the object detection as a regression problem and outputs the location and classification of the object on an end-to-end network in one step. The detection speed is one of the strengths of this method, but there is a large error in the detection accuracy of small targets [36]. The YOLO algorithm has been constantly improved and a simplified-structure version called YOLO v3-Tiny was developed. YOLO v3-Tiny uses a backbone network with only 7 convolutional layers and 6 pooling layers, and its FPN is also simplified by removing the maximum-scale prediction branch and decreasing the number of convolutional layers in the other two branches. In addition, YOLO v3-Tiny does not require a large amount of memory to be stored, reducing the need for hardware.

The test parameters used for the detection model are the following: a standard value of 0.5 for the IoU (intersection over union) threshold, and an NMS (non-maximum suppression) confidence threshold of 0.7. The aim of NMS [37] is to remove the prediction boxes whose overlap ratio is greater than a given threshold. To realize this, the algorithm selects the prediction box with the highest confidence to end up with only a unique bounding box, and the boxes whose IoUs are greater than a given threshold are discarded. The NMS algorithm is illustrated in Figure 5. When the confidence of the bounding box acquired is greater than the confidence threshold, it is considered a valid beluga candidate. In the final evaluation, when the overlapping area between the box predicted and the ground truth box is higher than the IoU threshold, this prediction is considered to be a correct prediction of the ground truth.

### 2.6. Evaluation Metrics for Object Detection

For measuring the performance of the detection algorithm, we rely on the mAP (mean average precision) metric, in which the AP is the integral of the PR (precision–recall) curve, which means the area under the PR curve. Precision and recall are also computed as described in Equations (Equation 1) and (Equation 2).
(1)$$P=\frac{T_{p}}{T_{p}+F_{p}}.$$
(2)$$R=\frac{T_{p}}{T_{p}+F_{n}}.$$

The precision (*P*) represents the correct detection rate of all predicted results, while the recall (*R*) corresponds to the correct detection rate with respect to all ground truths. In addition, $T_{p}$ (true positive) is the number of bounding boxes correctly predicted as containing a beluga, $F_{p}$ (false positive) is the number of bounding boxes inaccurately predicted as containing a beluga, and $F_{n}$ (false negative) is number of bounding boxes that are incorrectly predicted as containing a beluga. The line connecting the PR values determined by different confidence thresholds is the PR curve.

### 2.7. Training and Inference

Since these detectors have different training strategies, we described below in a separate manner how each of them were used. The Mask-RCNN and SSD models were first to be initialized with the parameters of the pre-trained model obtained using the COCO dataset [26], which adopts ResNet-50 as its backbone. More precisely, the models are trained during 50 epochs using a mini-batch of sixteen training images. For each epoch, image augmentation was applied for enhancing the model performance. This augmentation included operations such as random horizontal flips, brightness variation, and resizing. A total of 6 new images are obtained by rotating the original image 90, 180, 270, and 330 degrees clockwise, randomizing the brightness levels, and resizing the images between 0.0 and 1.0 [38]. Stochastic gradient descent [39] was used as the optimizer with the initial learning rate, momentum, and weight decay set, respectively, to 0.001, 0.9, and 0.0001. YOLO v3-Tiny adopts a DarkNet [25]-style backbone. Specifically, the model is trained during 50 epochs with an SGD optimizer and the same data augmentation used for Mask-RCNN and SSD. Here, the batch size is set to 16, the momentum to 0.9, and the decay to 0.0001.

During the inference phase for these three detector models (Mask-RCNN, SSD and YOLO v3-Tiny), we first resized the input images to 128 × 128. Second, we used pre-trained models, initializing weights with the COCO dataset [26]. Finally, we fine-tuned them using the Beluga-5k dataset. The detector backbone that we employed to perform without pre-trained weights is called the scratch model.

## 3. Results

As mentioned previously, we have conducted extensive experiments on the Beluga-5k dataset with three off-the-shelf object detectors, namely, Mask-RCNN, SSD, and YOLO v3-Tiny. Since our objective is to have a detection method displaying both high efficiency and effectiveness, we carefully compared these methods during preliminary experiments. As a result, YOLO v3-Tiny was clearly the method displaying the best trade-off.

### 3.1. Improving Manual with Semi-Manual Annotations

As previously discussed in Section 2.4, we firstly relied on manual annotations, and then performed semi-manual annotations to build the Beluga-5k dataset. To understand the improvement brought by the semi-manual annotation, compared to the manual one, we have trained a beluga detector on both annotations.

As shown in Table 2, the initial beluga detector trained with manual annotation achieves a reasonable detection performance of 85.23 (mAP@0.5) using validation subset, considering the intensive human labor on hundreds of images. Nonetheless, combining the semi-manual annotations with manual annotations for training greatly benefits the beluga detector, boosting the mAP by about 12.08%.

### 3.2. Off-the-Shelf Object Detectors for Belugas

Afterwards, we compared Mask-RCNN, SSD, and YOLO v3-Tiny on the Beluga-5k test images, with respect to detection performance and processing speed. As shown in Table 3, the slowest model is, in fact, Mask-RCNN. In addition, the Mask-RCNN and YOLO v3-Tiny perform much better than SSD in terms of mAP@0.5. A possible explanation of the poor accuracy of SSD is due to the “window mechanism”, which might not work well for small-scale objects. We have evaluated SSD grid cells at different layers. For instance, we have used a 4 × 4 and a 2 × 2 grid, but the central regions of the tiny box are too small to regress well. The average test time also indicates a slow processing speed.

Furthermore, we have observed that the speed of YOLO v3-Tiny is higher than Mask-RCNN under relatively similar accuracy. This is due to the fact that YOLO v3-Tiny adopts a lighter, more computational-friendly backbone (i.e., DarkNet) than that of Mask-RCNN (ResNet-50), and YOLO v3-Tiny also introduces extra-lightweight modules, including “Focus” and “CSP”, to refine the feature representations.

### 3.3. Performance of Beluga Detection under the Pre-Trained and Scratch Models

The performance of the YOLO v3-Tiny detector on test images applying the pre-trained and scratch backbone, as mentioned in Section 2.7, is shown in Figure 6. Following the order used in the legend, the AP values of these methods are, respectively, 95.80, 97.05, 89.11, and 95.23.

The results indicate that a pre-trained model has a stronger detection ability. Indeed, the AP value of the pre-trained model is higher than the scratch model (by 6.69 and 1.82 for the manual and semi-manual annotations, respectively). The pre-trained model learns a large number of object features from the COCO dataset [26], with 80 classes and more than 1.5 million labeled objects. Afterwards, when the model has to deal with beluga individuals after fine-tuning, it will display a high discrimination power. This improves the detection performance, which is enhanced by the transfer learning that has occurred.

### 3.4. Detection on Images

Three of the detected beluga instances corresponding to the pre-trained and scratch approaches can be visualized in Figure 7. These images are challenging to detect due to difficult backgrounds, low resolutions, and foam distractors. We observed that training the YOLO v3-Tiny detector from scratch using the manual approach is not effective, as is shown in Figure 7A,E,I, due to the insufficient number of annotations. We have increased the number of annotations using the semi-manual approach with mixed samples (i.e., single- and multiple-beluga images), which has led to a small enhancement of the detector, as seen in Figure 7F,J, except for Figure 7B. In these examples, even if the pre-trained model is still not able to detect difficult backgrounds (Figure 7C), it has detected with success the low-resolution (Figure 7G) and the foam distractor (Figure 7K). Finally, the difficult samples can be accurately detected (Figure 7D,H,L) when a pre-trained YOLO v3-Tiny model with semi-manual annotations is used. To summarize, our best detection model is obtained by (1) using mixture-annotations images (single- and multiple-individual), (2) increasing the number of annotations, and (3) using pre-trained CNN weights. This means that, in our setting, a combination of transfer learning and fine-tuning can significantly improve the detection performance.

The model obtained is also able to detect multiple objects in an image, as shown in Figure 8. The same approaches were also investigated with multiple-beluga samples, which led to the same conclusion—with the exception that the manual approach had even worse results than previously presented.

Some false alarms from the manual pre-trained model occurred due to distractor objects such as occlusions/overlaps among belugas. Figure 8A shows a misdetection in which a wave as been labelled as a small beluga. Similarly, an error in the detection is seen in Figure 8D,J. Figure 8G displays a situation in which two beluga candidates were detected as a single one. Since the manual approach is composed of only single annotations, it causes incomplete detection when belugas are close to each other. In contrast, the semi-manual approach is trained using mixed annotations, resulting in the the detector unveiling occlusions caused by distractors or close individuals (Figure 8B,E,K). However, overlapped belugas remain undetectable in Figure 8H. Our best detector model, which has been applied in the last column’s images (Figure 8C,F,I,L), increases the detection power by leveraging the pre-trained backbone model on the YOLO v3-Tiny detector, using mixed-dataset annotations. Those annotations are better for extracting discriminative details from multiple-individual images, even in images in which there are challenging overlaps among objects (Figure 8I).

### 3.5. Case Studies for Single- and Multiple-Beluga Images

In this section, we report our results on successful and unsuccessful detection for single- and multiple-beluga images. As seen in Table 4, the proposed method achieves a precision of 99.94% and a recall of 99.87% for single-individual images, and a precision of 98.29% and a recall of 99.14% for multiple-individual images.

A successful case study of our proposed beluga-detection method is presented in Figure 9. However, it should be noted that false negatives of beluga detection can occur due to an occlusion by a D-Tag (Figure 10A). False positives are sometimes due to items on the water that were detected as a beluga (e.g., shadow, reflection, and wave objects in Figure 10B,E,D, respectively). For example, the shape of the false-positive object highly resembled a true-positive (TP) object in Figure 10E,D. Nevertheless, our method may present these false-positive objects into single cropped images, as seen in Figure 11, to be filtered out by specialists. Sometimes, some unsuccessful detections can occur in multiple images. For instance, Figure 10C shows the same beluga detected as two beluga candidates, or does not recognize the single beluga, as demonstrated in Figure 10F.

Finally, Figure 11 illustrates the output image result obtained by our method on a image containing five belugas. More precisely, for each test image (single- or multiple-individual), our detection method outputs their respective cropped images after detecting them. The cropped output removes the complex ocean background, which we believe will increase the performance of the next step of the project, which is the classification of individuals.

## 4. Discussion

The experimental results show that the proposed method is an effective and noninvasive way to detect beluga whales in the ocean. When comparing our results to previous research, it is important to consider the type of data being used. As far as we know, we are the first work to recognize images with multiple beluga whales; these images are formed by whales swimming in groups of which overlap, and occlusion becomes a non-trivial task in efficient detection. For instance, Harasyn et al. [27] designed a beluga-detection approach using a small dataset with 721 beluga images extracted from drone videos. Despite the fact that images were captured by drones, most of the photos contain small-scale belugas surrounded by a huge background sea. Moreover, the detection of multiple belugas in the same picture is a challenging task that has not been addressed by the authors.

We have trained state-of-the-art object-detection algorithms based on the Beluga-5k dataset and compared their performance. Based on these results, the YOLO v3-Tiny algorithm is shown to outperform the others, both in terms of speed and prediction accuracy. Our fine-tuned CNN-backbone detector trained with semi-manual annotations has a stronger detection ability than the scratch model with manual annotation. Indeed, the AP value of the pre-trained model is higher than the scratch model (by 6.69 and 1.82 for the manual and semi-manual annotations, respectively). The semi-manual annotation benefits the beluga detector in two ways. First, once the manual annotations are defined, the next annotations are automatically made. Thus, a large-scale annotation could be performed by training the detector model without intensive human labor. Second, single- and multiple-beluga annotations are trained together. We speculate that this approach helps the detector in understanding occlusions and overlaps in complex images. In addition, we evaluated the possibility to detect multiple beluga whales. In particular, 200 test images were evaluated with a precision of 98.29% and a recall of 99.14%. As seen in Figure 11, the proposed detection strategy is useful for images containing multiple belugas, as it can separate beluga individuals as unique instances and then automatically separate them or submit them to the photographic survey.

Looking at the detected images in Figure 10, the main related errors are distractor objects (e.g., waves, sun glare, and low-resolution photos). We consider low resolution as a distractor for older images, although we acknowledge that currently the problem of low resolution can be solved by improving the quality of the camera. We can observe that, in comparison to large sea backgrounds, belugas are small on these images, and specialists’ manual efforts require detailed inspections. In addition, human whale detectors are capable of detecting belugas through visual clues (e.g., dorsal ridged, scars, and shape) in these large and complicated backgrounds. Thus, automatically finding belugas in the wild is still as challenging as visual sightings of wild belugas are (very small, low resolution, overlapped, occluded, etc.).

## 5. Conclusions

In this paper, we designed a semi-automated detection framework to detect the beluga whale population in the wild. To achieve this, we first developed a two-step (one manual and one semi-manual step) annotation technique to create our labeled dataset, Beluga-5k. Afterwards, we conducted experiments to compare different off-the-shelf detection methods before fine-tuning our approach to the data at hand, and achieved high performance in terms of accuracy as well as processing speed with tiny-YOLO v3. The results indicate that a fine-tuned model has a stronger detection ability; also, the AP value of the pre-trained model is higher than the scratch model (by 6.69 and 1.8 for the manual and semi-manual annotations, respectively). Moreover, the same proposed method for single belugas can be easily used to detect multiple belugas with occlusion/overlap complexity in the image. We believe that the proposed framework can assist marine biologists and researchers in cataloging the beluga whale population and developing conservation strategies without manual effort. In future works, we plan to integrate our detection approach as a pre-processing step for a classification system. The use of metric models such as Siamese and triplet networks will be evaluated to deal with open-set dataset challenges. To realize this, traditional neural network architectures such as ResNet and DenseNet will be used to extract the salient features of beluga whales that are similar to each other.

## Figures and Tables

**Figure 1 sensors-22-04107-f001:**
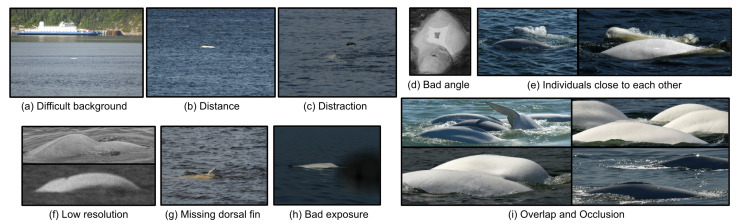
Examples of image distracters that either lead to completely unusable data samples or make a robust and correct detection much more complicated (false alarm).

**Figure 2 sensors-22-04107-f002:**
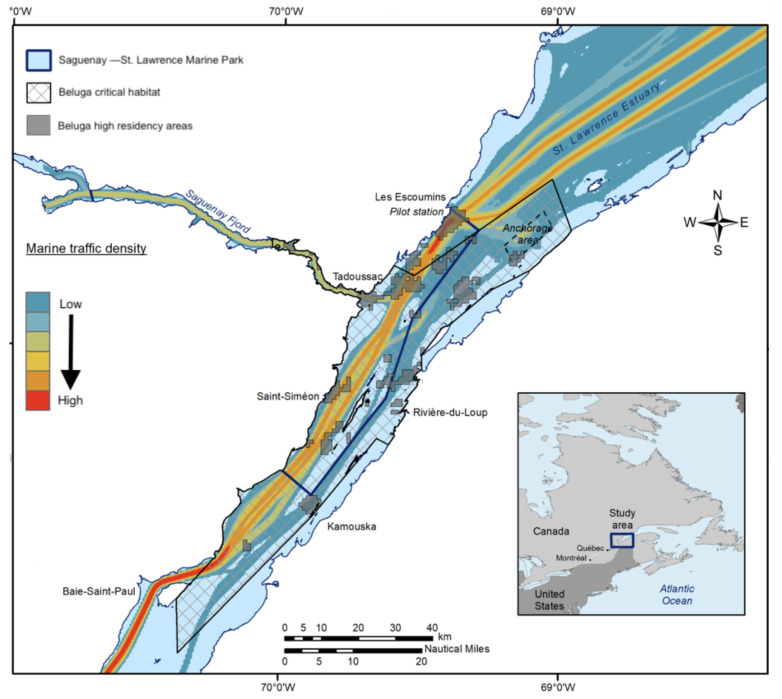
Study area. The photo-identification campaign is mostly conducted within the critical habitat of the St. Lawrence Estuary Beluga population in and around the Saguenay–St. Lawrence Marine Park.

**Figure 3 sensors-22-04107-f003:**
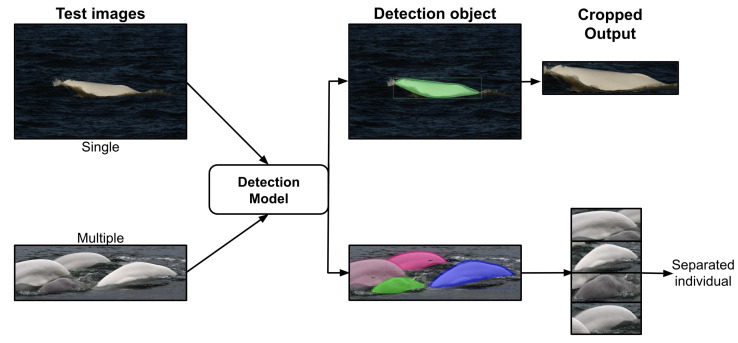
Overview of our proposed method.

**Figure 4 sensors-22-04107-f004:**
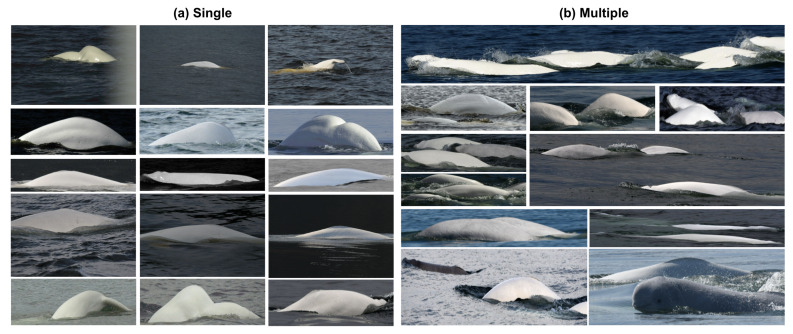
(**a**) Single: contains one beluga individual per image; (**b**) multiple: contains two or more beluga individuals per image.

**Figure 5 sensors-22-04107-f005:**
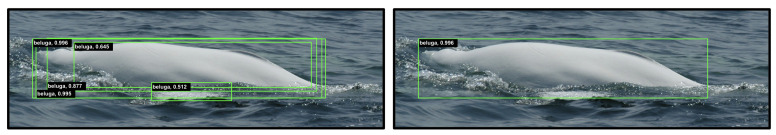
Before and after applying the non-maximum suppression (NMs) algorithm.

**Figure 6 sensors-22-04107-f006:**
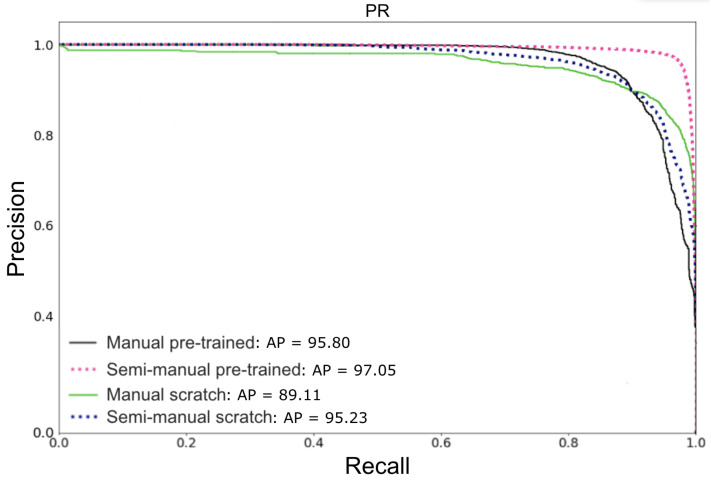
PR curves and AP values of experiments using manual and semi-manual annotation datasets, applying the pre-trained and scratch YOLO v3-Tiny detector.

**Figure 7 sensors-22-04107-f007:**
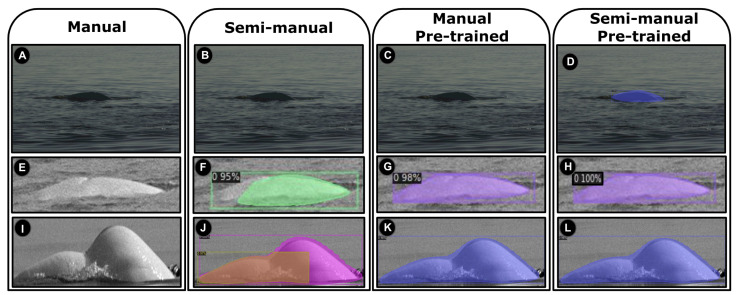
Scratch and pre-trained detection models are performed on single-beluga test images. Images (**A**,**E**,**I**) receive no detection due to an insufficient number of manual annotations; (**B**,**C**) are without detection due to their complex backgrounds; (**F**,**J**) confuse detection; (**G**,**K**) improve detections using pre-trained manual annotations; (**D**,**H**,**L**) has successful detections due to the application of the pre-trained model with semi-manual annotations. The detected objects are represented by colorful rectangles.

**Figure 8 sensors-22-04107-f008:**
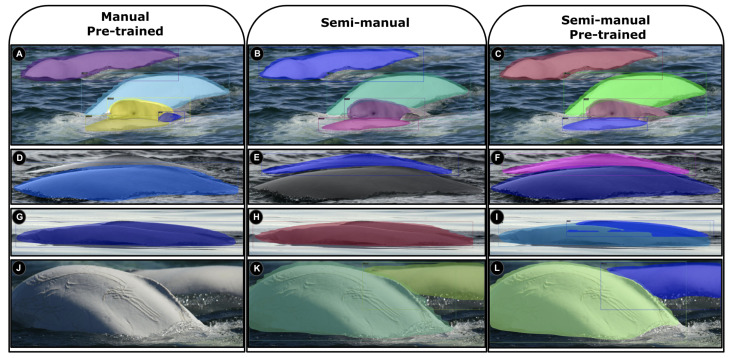
Scratch and pre-trained detection models are applied on multiple-beluga test images. Each line represents a beluga whale sample. Image (**A**) has a misdetection due to a wave being considered a small beluga; (**B**,**C**) correctly detect the 4 belugas in the image; (**D**) has a misdetection in occluded scenarios; (**E**,**F**) have an improved detection in cases of occlusion by using semi-manual annotations and pre-trained model; (**G**,**H**) display a situation in which two beluga candidates were detected as a single one; (**I**) corresponds to a successful detection between two nearby belugas; (**J**) has no detection; (**K**,**L**) have an improved detection by using semi-manual annotations and pre-trained model. The detected objects are represented by colorful rectangles.

**Figure 9 sensors-22-04107-f009:**
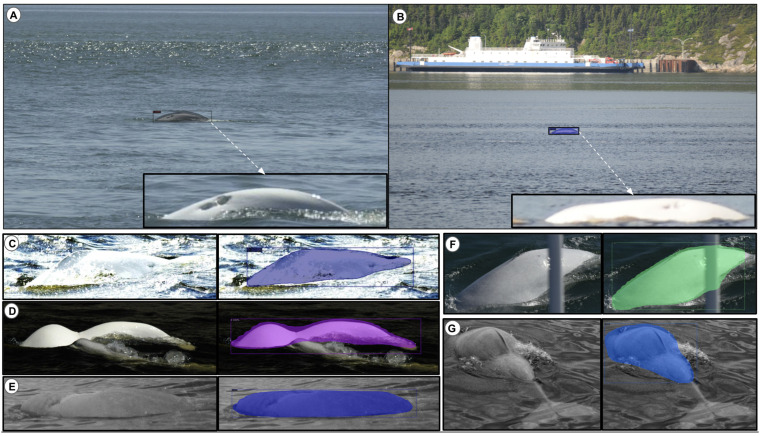
Successful detection in complex scenarios (presence of distractor objects): (**A**) distance; (**B**) difficult background with distance; (**C**) bad exposure; (**D**) belugas close to each other; (**E**) low-resolution image; (**F**) unexpected distractor; (**G**) bad orientation of dorsal ridge.

**Figure 10 sensors-22-04107-f010:**
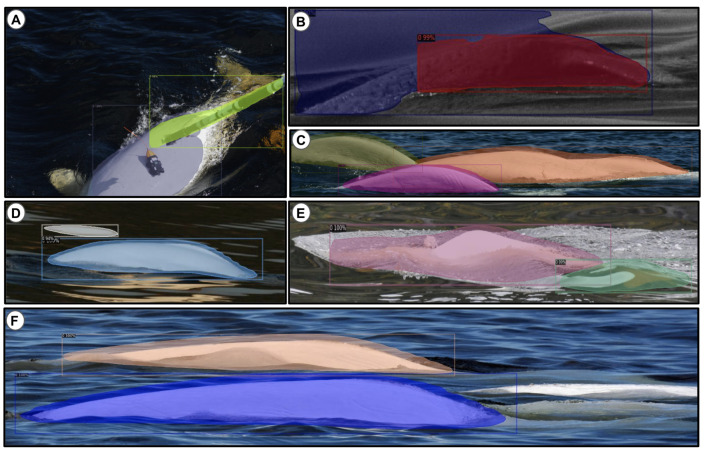
Failed detection due to different causes: (**A**) false-negative detections; (**B**,**D**,**E**) false-positive detections; (**C**) two beluga candidates occluded by water, misdetected as one beluga; (**F**) a missing dorsal ridge.

**Figure 11 sensors-22-04107-f011:**
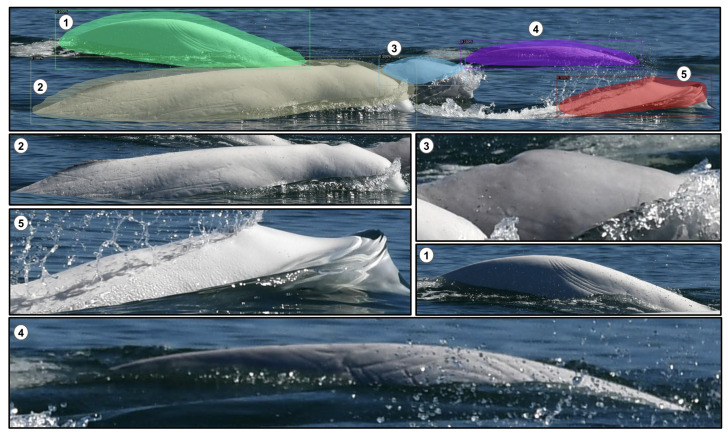
Cropping beluga individuals in the multiple-beluga images. Five belugas (numbers 1 to 5) are detected in the first image (first line). Each following cropped image with a number represents the individual associated with the first image.

**Table 1 sensors-22-04107-t001:** Number of individuals and training, validation, and test images, as well as candidate belugas, for single- and multiple-individual images.

		Number of				
Image	Individuals	Training Images	Validation Images	Test Images	Candidates	Average Number of Candidates per Image
Single	427	3182	100	1590	4872	1
Multiple	-	50	0	200	706	3.5
Total	427	3232	100	1790	5578	-

**Table 2 sensors-22-04107-t002:** Impacts of the manual and semi-manual annotation processes on detection performance mAP@0.5 (%).

Bakcbone	Manual	Semi-Manual
YOLO v3-Tiny	85.23	96.31

**Table 3 sensors-22-04107-t003:** Comparisons of three object detectors: Mask-RCNN, SSD, and YOLO v3-Tiny. The average time (ms) is computed considering 30 images processed in each model.

Bakcbone	mAP@0.5 (%)	Average Time (ms)
Mask-RCNN	98.80	542.1
SSD	87.82	383.2
YOLO v3-Tiny	97.05	135.2

**Table 4 sensors-22-04107-t004:** Analysis of the proposed beluga-detection method.

Beluga	Test Images	Ground Truth Candidates	TP	FP	FN	Precision (%)	Recall (%)
Single	1590	1590	1587	1	2	99.94	99.87
Multiple	200	706	688	12	6	98.29	99.14

## Data Availability

Not applicable.

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
