# Peer review of "Machine-Learning Approach for Automatic Detection of Wild Beluga Whales from Hand-Held Camera Pictures"

_sensors, 2022, doi:10.3390/s22114107_

Round 1

Reviewer 1 Report

See attached file

Author Response

The answers are in the attached pdf document.

Reviewer 2 Report

This paper focuses on the detection and recognition of wild beluga whales from hand-held camera pictures.
From the perspective of computer science (CS), this paper is quite weak in technical innovation, because almost all key technical components are the off-the-shelf ones, and not much new can be found.
However, in the view of the field of ocean protection, this paper really deserves a positive recommendation, where the engineering related details are quite solid and persuasive.
Besides, this paper has multiple merits, e.g., good paper writing, reasonable paper structure, and clear pictorial illustrations.
And the newly proposed dataset is another highlight.
Despite the above mentioned merits, I still have several concerns, which could be help the authors to improve the paper further.

1) As shown in Fig. 1, the authors have clearly stated that the proposed approach can handle the "unknown" category, yet, in the main content, nothing related with this issue can be found. It just seems like that the authors have completely overlooked the claimed re-identification task.

2) In the introduction part, the authors have summarized multiple challenges towards the beluga whales detection task, i.e., line 53-69. However, in my view, the last two points are just the common ones, and thus they should be removed.

3) Line 70-83 should be removed to the related work section, since these contents are clearly talking about the backgrounds. 

4) Though this paper have constructed a new dataset, I cannot find any web link to access it. To me, this paper can really benefit our research community only if the claimed new set is made publicly available.

5) In sec. 3.3, the proposed semi-annotation approach is quite similar to several existing works [1], and thus the authors should provide a brief discussion towards the main differences.
[1] A Novel Video Salient Object Detection Method via Semisupervised Motion Quality Perception;

6) In sec. 4.2, the authors have adopted a pre-defined hard threshold to remove false-alarms. However, this implementation will leads to a trade-off between precision and recall. To better ground this issue, I think that an additional ablation study with in-depth discussion should be provided.

7) The counting related experiment is less persuasive, i.e., Table 4. For images containing a single beluga, the counting task is almost meaningless. 
Though I feel generally happy with this paper, a careful revision is really necessary. Thus. I think that this paper should undergo a major revision.

Author Response

(The authors gave the same response as above.)

Round 2

Reviewer 2 Report

Most of my previous concerns have addressed, and thus I think that this paper is ready for publication.